# MULTI-TURN DIALOGUE RESPONSE GENERATION IN AN ADVERSARIAL LEARNING FRAMEWORK

## ABSTRACT

We propose an adversarial learning approach to the generation of multi-turn dialogue responses. Our proposed framework, *hredGAN*, is based on conditional generative adversarial networks (GANs). The GAN's generator is a modified hierarchical recurrent encoder-decoder network (HRED) and the discriminator is a word-level bidirectional RNN that shares context and word embedding with the generator. During inference, noise samples conditioned on the dialogue history are used to perturb the generator's latent space to generate several possible responses. The final response is the one ranked best by the discriminator. The hredGAN shows major advantages over existing methods: (1) it generalizes better than networks trained using only the log-likelihood criterion, and (2) it generates longer, more informative and more diverse responses with high utterance and topic relevance even with limited training data. This superiority is demonstrated on the Movie triples and Ubuntu dialogue datasets with both the automatic and human evaluations.

## 1 INTRODUCTION

Recent advances in deep neural network architectures have enabled tremendous success on a number of difficult machine learning problems. While these results are impressive, producing a deployable neural network–based conversation model that can engage in open domain discussion still remains elusive. A dialogue system needs to be able to generate meaningful and diverse responses that are simultaneously coherent with the input utterance and the overall dialogue topic. Unfortunately, earlier conversation models trained with naturalistic dialogue data suffered greatly from limited contextual information (Sutskever et al., 2014; Vinyals & Le, 2015), and lack diversity (Li et al., 2016a). These problems often leads to generic and safe utterance in response to varieties of input utterance.

Serban et al. (2016) and Xing et al. (2017) proposed the Hierarchical Recurrent Encoder-Decoder (HRED) network to capture long temporal dependencies in multi-turn conversations to address the limited contextual information but the diversity problem remained. On the other hand, some HRED variants such as variational (Serban et al., 2017b) and multi-resolution (Serban et al., 2017a) HREDs attempt to alleviate the diversity problem by injecting noise at the utterance level and by extracting additional context to condition the generator on. While these approaches achieve certain measures of success over the basic HRED, generated responses are still mostly generic since they do not control the generator's output as the output conditional distribution is not calibrated. Li et al. (2016a), on the other hand, consider diversity promoting training objective but their model is for single turn conversations, cannot not be trained end-to-end and therefore achieves little.

The generative adversarial network (GAN) (Goodfellow et al., 2014) seems to be an appropriate solution to the diversity problem. GAN matches data from two different distributions by introducing an adversarial game between a *generator* and a *discriminator*. We explore *hredGAN*: conditional GANs for multi-turn dialogue models with HRED generator and discriminator. hredGAN combines both generative and retrieval-based multi-turn dialogue systems to improve their individual performances. This is achieved by sharing the context and word embedding between the generator and the discriminator allowing for joint end-to-end training using back-propagation. To the best of our knowledge, no existing work has applied conditional GANs to multi-turn dialogue models and especially with HRED generators and discriminators. We demonstrate the effectiveness of hredGAN

over the VHRED for dialogue modeling with evaluations on the Movie triples and Ubuntu technical support datasets.

## 2 ADVERSARIAL FRAMEWORK FOR MULTI-TURN DIALOGUE

Consider a dialogue consisting of a sequence of $N$ utterances, $\boldsymbol{X} = (X_1, X_2, \cdots, X_N)$, where each utterance $X_i = (X_i^1, X_i^2, \cdots, X_i^{M_i})$ contains a variable-length sequence of $M_i$ word tokens such that $X_i^j \in V$ for vocabulary $V$. At any time step $i$, the dialogue history is given by $\boldsymbol{X_i} = (X_1, X_2, \cdots, X_i)$. The dialogue response generation task can be defined as follows: Given a dialogue history $\boldsymbol{X_i}$, generate a response $Y_i = (Y_i^1, Y_i^2, \cdots, Y_i^{T_i})$, where $T_i$ is the number of generated tokens. We also want the distribution of the generated response $P(Y_i)$ to be indistinguishable from that of the ground truth $P(X_{i+1})$ and $T_i = M_{i+1}$. Conditional GAN learns a mapping from an observed dialogue history, $\boldsymbol{X_i}$, and a sequence of random noise vectors, $Z_i$ to a sequence of output tokens, $Y_i$, $G : \{\boldsymbol{X_i}, Z_i\} \rightarrow Y_i$. The generator $G$ is trained to produce output sequences that cannot be distinguished from the ground truth sequence by an adversarially trained discriminator $D$ that is trained to do well at detecting generator's fakes. The distribution of the generator output sequence can be factored by the product rule:

$$P(Y_i|\boldsymbol{X_i}) = P(Y_i^1)\prod_{j=2}^{T_i} P(Y_i^j|Y_i^1, \cdots, Y_i^{j-1}, \boldsymbol{X_i}) \tag{1}$$

$$P(Y_i^j|Y_i^1, \cdots, Y_i^{j-1}, \boldsymbol{X_i}) = P_{\theta_G}(Y_i^{1:j-1}, \boldsymbol{X_i}) \tag{2}$$

where $Y_i^{i:j-1} = (Y_i^1, \cdots, Y_i^{j-1})$ and $\theta_G$ are the parameters of the generator model. $P_{\theta_G}(Y_i^{i:j-1}, \boldsymbol{X_i})$ is an autoregressive generative model where the probability of the current token depends on the past generated sequence. Training the generator $G$ with the log-likelihood criterion is unstable in practice, and therefore the past generated sequence is substituted with the ground truth, a method known as *teacher forcing* (Williams & Zipser, 1989), i.e.,

$$P(Y_i^j|Y_i^1, \cdots, Y_i^{j-1}, \boldsymbol{X_i}) \approx P_{\theta_G}(X_{i+1}^{1:j-1}, \boldsymbol{X_i}) \tag{3}$$

Using equation 3 in relation to GAN, we define our fake sample as the teacher forcing output with some input noise $Z_i$

$$Y_i^j \sim P_{\theta_G}(X_{i+1}^{1:j-1}, \boldsymbol{X_i}, Z_i) \tag{4}$$

and the corresponding real sample as ground truth $X_{i+1}^j$.

With the GAN objective, we can match the noise distribution, $P(Z_i)$ to the distribution of the ground truth response, $P(X_{i+1}|\boldsymbol{X_i})$. Varying the noise input then allows us to generate diverse responses to the same dialogue history. Furthermore, the discriminator, since it is calibrated, is used during inference to rank the generated responses, providing a means of controlling the generator output.

### 2.1 OBJECTIVES

The objective of a conditional GAN can be expressed as

$$\mathcal{L}_{cGAN}(G, D) = \mathbb{E}_{\boldsymbol{X_i}, X_{i+1}}[\log D(X_{i+1}, \boldsymbol{X_i})] + \mathbb{E}_{\boldsymbol{X_i}, Z_i}[1 - \log D(G(\boldsymbol{X_i}, Z_i), \boldsymbol{X_i})] \tag{5}$$

where $G$ tries to minimize this objective against an adversarial $D$ that tries to maximize it:

$$G^*, D^* = arg \min_G \max_D \mathcal{L}_{cGAN}(G, D). \tag{6}$$

Previous approaches have shown that it is beneficial to mix the GAN objective with a more traditional loss such as cross-entropy loss (Lamb et al., 2016; Li et al., 2017). The discriminator's job remains unchanged, but the generator is tasked not only to fool the discriminator but also to be near the ground truth $X_{i+1}$ in the cross-entropy sense:

$$\mathcal{L}_{MLE}(G) = \mathbb{E}_{\boldsymbol{X_i}, X_{i+1}, Z_i}[-log P_{\theta_G}(X_{i+1}, \boldsymbol{X_i}, Z_i)]. \tag{7}$$

Our final objective is,

$$G^*, D^* = arg \min_G \max_D (\lambda_G \mathcal{L}_{cGAN}(G, D) + \lambda_M \mathcal{L}_{MLE}(G)). \tag{8}$$

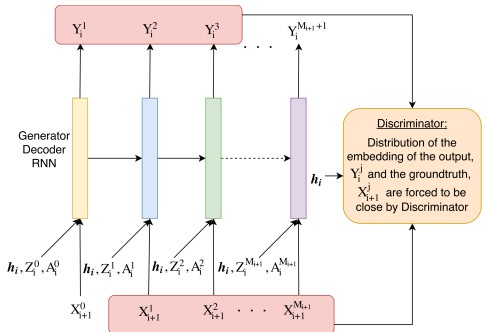 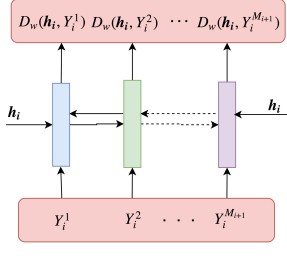

Figure 1: **Left: The hredGAN architecture -** The generator makes predictions conditioned on the dialogue history, $\boldsymbol{h_i}$, attention, $A_i^j$, noise sample, $Z_i^j$, and ground truth, $X_{i+1}^{j-1}$. **Right: RNN-based discriminator** that discriminates bidirectionally at the word level.

It is worth mentioning that, without $Z_i$, the net could still learn a mapping from $\boldsymbol{X_i}$ to $Y_i$, but would produce deterministic outputs and fail to match any distribution other than a delta function (Isola et al., 2017). This is one key area where our work is different from Lamb et al.'s and Li et al.'s. The schematic of the proposed hredGAN is depicted at the right hand side of Figure 1.

## 2.2 GENERATOR

We adopted an HRED dialogue generator similar to (Serban et al., 2016; 2017a;b; Xing et al., 2017). The HRED contains three recurrent structures, i.e. the encoder ($eRNN$), context ($cRNN$), and decoder ($dRNN$) RNN. The conditional probability modeled by the HRED per output word token is given by

$$P_{\theta_G}\big(Y_i^j|X_{i+1}^{1:j-1}, \boldsymbol{X_i}\big) = dRNN\big(E(X_{i+1}^{j-1}), h_i^{j-1}, \boldsymbol{h_i}\big) \tag{9}$$

where $E(.)$ is the embedding lookup, $\boldsymbol{h_i} = cRNN(eRNN(E(X_i)), \boldsymbol{h_{i-1}})$, $eRNN(.)$ maps a sequence of input symbols into fixed-length vector, and $h$ and $\boldsymbol{h}$ are the hidden states of the decoder and context RNN, respectively.

In the multi-resolution HRED, (Serban et al., 2017a), high-level tokens are extracted and processed by another RNN to improve performance. We circumvent the need for this extra processing by allowing the decoder to attend to different parts of the input utterance during response generation (Bahdanau et al., 2015; Luong et al., 2015). We introduce a local attention into equation 9 and encode the attention memory differently from the context through an attention encoder RNN ($aRNN$), yielding:

$$P_{\theta_G}\big(Y_i^j|X_{i+1}^{1:j-1}, \boldsymbol{X_i}\big) = dRNN\big(E(X_{i+1}^{j-1}), h_i^{j-1}, A_i^j, \boldsymbol{h_i}\big) \tag{10}$$

where $A_i^j = \sum_{m=1}^{M_i} \frac{exp(\alpha_m)}{\sum_{m=1}^{M_i} exp(\alpha_m)} h_i^{'m}$, $h_i^{'m} = aRNN(E(X_i^m), h_i^{'m-1})$, $h^{'}$ is the hidden state of the attention RNN and $\alpha_k$ is either a logit projection of $(h_i^{j-1}, h_i^{'m})$ in the case of Bahdanau et al. (2015) or $(h_i^{j-1})^T \cdot h_i^{'m}$ in the case of Luong et al. (2015). The modified HRED architecture is shown in Figure 2.

**Noise Injection:** We inject Gaussian noise at the input of the decoder RNN. Noise samples could be injected at the utterance or word level. With noise injection, the conditional probability of the decoder output becomes

$$P_{\theta_G}\big(Y_i^j|X_{i+1}^{1:j-1}, Z_i^j, \boldsymbol{X_i}\big) = dRNN\big(E(X_{i+1}^{j-1}), h_i^{j-1}, A_i^j, Z_i^j, \boldsymbol{h_i}\big) \tag{11}$$

where $Z_i^j \sim \mathcal{N}_i(0, \boldsymbol{I})$, for utterance-level noise and $Z_i^j \sim \mathcal{N}_i^j(0, \boldsymbol{I})$, for word-level noise.

## 2.3 DISCRIMINATOR

The discriminator shares context and word embedding with the generator and can discriminate at the word level (Lamb et al., 2016). The word-level discrimination is achieved through a bidirectional RNN and is able to capture both syntactic and conceptual differences between the generator output

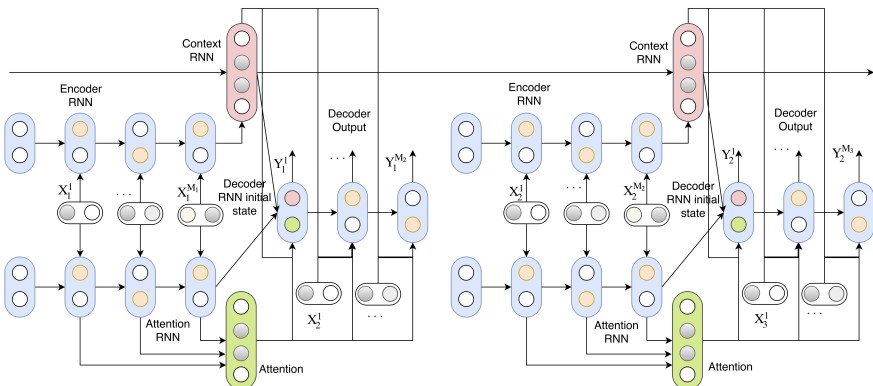

Figure 2: **The HRED generator with local attention -** The attention RNN ensures local relevance while the context RNN ensures global relevance. Their states are combined to initialize the decoder RNN and the discriminator BiRNN.

and the ground truth. The aggregate classification of an input sequence, $\chi$ can be factored over word-level discrimination and expressed as

$$D(\boldsymbol{X_i}, \chi) = D(\boldsymbol{h_i}, \chi) = \left[ \prod_{j=1}^{J} D_{RNN}(\boldsymbol{h_i}, E(\chi^j)) \right]^{\frac{1}{J}} \qquad (12)$$

where $D_{RNN}(.)$ is the word discriminator RNN, $\boldsymbol{h_i}$ is an encoded vector of the dialogue history $\boldsymbol{X_i}$ obtained from the generator's $cRNN(.)$ output, and $\chi^j$ is the *jth* word or token of the input sequence $\chi$. $\chi = Y_i$ and $J = T_i$ for the case of generator's decoder output, $\chi = X_{i+1}$ and $J = M_{i+1}$ for the case of ground truth. The discriminator architecture is depicted on the left hand side of Figure 1.

## 2.4 ADVERSARIAL GENERATION OF MULTI-TURN DIALOGUE RESPONSE

In this section, we describe the generation process during inference. The generation objective can be mathematically described as

$$Y_i^* = arg\max_l \left\{ P(Y_{i,l} | \boldsymbol{X_i}) + D^*(\boldsymbol{X_i}, Y_{i,l}) \right] \right\}_{l=1}^L \qquad (13)$$

where $Y_{i,l} = G^*(\boldsymbol{X_i}, Z_{i,l})$, $Z_{i,l}$ is the *lth* noise samples at dialogue step $i$, and $L$ is the number of response samples. Equation 13 shows that our inference objective is the same as the training objective (8), combining both the MLE and adversarial criteria. This is in contrast to existing work where the discriminator is usually discarded during inference.

The inference described by equation 13 is intractable due to the enormous search space of $Y_{i,l}$. Therefore, we turn to an approximate solution where we use greedy decoding (MLE) on the first part of the objective function to generate $L$ lists of responses based on noise samples $\{Z_{i,l}\}_{l=1}^L$. In order to facilitate the exploration of the generator's latent space, we sample a modified noise distribution, $Z_{i,l}^j \sim \mathcal{N}_{i,l}(0, \alpha\boldsymbol{I})$, or $Z_{i,l}^j \sim \mathcal{N}_{i,l}^j(0, \alpha\boldsymbol{I})$ where $\alpha > 1.0$, is the exploration factor that increases the noise variance. We then rank the $L$ lists using the discriminator score, $\left\{ D^*(\boldsymbol{X_i}, Y_{i,l}) \right] \right\}_{l=1}^L$. The response with the highest discriminator ranking is the optimum response for the dialogue context.

## 3 TRAINING OF HREDGAN

We trained both the generator and the discriminator simultaneously as highlighted in Algorithm 1 with $\lambda_G = \lambda_M = 1$. GAN training is prone to instability due to competition between the generator and the discriminator. Therefore, parameter updates are conditioned on the discriminator performance (Lamb et al., 2016).

**The generator** consists of four RNNs with different parameters, that is, $aRNN, eRNN, cRNN,$ and $dRNN$. $aRNN$ and $eRNN$ are both bidirectional, while $cRNN$ and $dRNN$ are unidirec-

---

**Algorithm 1** Adversarial Learning of hredGAN

---

**Require:** A generator $G$ with parameters $\theta_G$.
**Require:** A discriminator $D$ with parameters $\theta_D$.
  **for** number of training iterations **do**
    Initialize $cRNN$ to zero_state, $\boldsymbol{h_0}$
    Sample a mini-batch of conversations, $\boldsymbol{X} = \{X_i\}_{i=1}^N$, $\boldsymbol{X_i} = (X_1, X_2, \cdots, X_i)$ with $N$ utterances. Each utterance mini batch $i$
    contains $M_i$ word tokens.
    **for** $i = 1$ **to** $N - 1$ **do**
      Update the context state.
      $\boldsymbol{h_i} = cRNN(eRNN(E(X_i)), \boldsymbol{h_{i-1}})$
      Compute the generator output using equation 11.
      $P_{\theta_G}\left(Y_i|, Z_i, \boldsymbol{X_i}\right) = \left\{ P_{\theta_G}\left(Y_i^j | X_{i+1}^{1:j-1}, Z_i^j, \boldsymbol{X_i}\right) \right\}_{j=1}^{M_i+1}$
      Sample a corresponding mini batch of utterance $Y_i$.
      $Y_i \sim P_{\theta_G}\left(Y_i|, Z_i, \boldsymbol{X_i}\right)$
    **end for**
    Compute the discriminator accuracy $D_{acc}$ over $N - 1$ utterances $\{Y_i\}_{i=1}^{N-1}$ and $\{X_{i+1}\}_{i=1}^{N-1}$
    **if** $D_{acc} < acc_{D_{th}}$ **then**
      Update $\theta_D$ with gradient of the discriminator loss.
      $\sum_i [\nabla_{\theta_D} \log D(\boldsymbol{h_i}, X_{i+1}) + \nabla_{\theta_D} log\left(1 - D(\boldsymbol{h_i}, Y_i)\right)]$
    **end if**
    **if** $D_{acc} < acc_{G_{th}}$ **then**
      Update $\theta_G$ with the generator's MLE loss only.
      $\sum_i [\nabla_{\theta_G} \log P_{\theta_G}\left(Y_i|, Z_i, \boldsymbol{X_i}\right)]$
    **else**
      Update $\theta_G$ with both adversarial and MLE losses.
      $\sum_i [\lambda_G \nabla_{\theta_G} \log D(\boldsymbol{h_i}, Y_i) + \lambda_M \nabla_{\theta_G} \log P_{\theta_G}\left(Y_i|, Z_i, \boldsymbol{X_i}\right)]$
    **end if**
  **end for**

---

tional. Each RNN has 3 layers, and the hidden state size is 512. The $dRNN$ and $aRNN$ are connected using an additive attention mechanism (Bahdanau et al., 2015).

**The discriminator** shares $aRNN, eRNN$, and $cRNN$ with the generator. $D_{RNN}$, is a stacked bidirectional RNN with 3 layers and a hidden state size of 512. The $cRNN$ states are used to initialize the states of $D_{RNN}$. The output of both the forward and the backward cells for each word are concatenated and passed to a fully-connected layer with binary output. The output is the probability that the word is from the ground truth given the past and future words of the sequence.

**Others:** All RNNs used are gated recurrent unit (GRU) cells (Cho et al., 2014). The word embedding size is 512 and shared between the generator and the discriminator. The initial learning rate is $0.5$ with decay rate factor of $0.99$, applied when the adversarial loss has increased over two iterations. We use a batch size of 64 and clip gradients around $5.0$. As in Lamb et al. (2016), we find $acc_{D_{th}} = 0.99$ and $acc_{G_{th}} = 0.75$ to be good enough. All parameters are initialized with Xavier uniform random initialization (Glorot & Bengio, 2010). The vocabulary size $V$ is $50,000$. Due to the large vocabulary size, we use sampled softmax loss (Jean et al., 2015) for MLE loss to expedite the training process. However, we use full softmax for evaluation. The model is trained end-to-end using the stochastic gradient descent algorithm.

## 4 EXPERIMENTS AND RESULTS

We consider the task of generating dialogue responses conditioned on the dialogue history and the current input utterance. We compare the proposed hredGAN model against some alternatives on publicly available datasets.

### 4.1 DATASETS

**Movie Triples Corpus**, (MTC) dataset (Serban et al., 2016). This dataset was derived from the *Movie-DiC* dataset by Banchs (2012). Although this dataset spans a wide range of topics with few spelling mistakes, its small size of only about 240,000 dialogue triples makes it difficult to train a dialogue model, as pointed out by Serban et al. (2016). We thought that this scenario would really benefit from the proposed adversarial generation.

**Ubuntu Dialogue Corpus**, (UDC) dataset (Serban et al., 2017b). This dataset was extracted from the Ubuntu Relay Chat Channel. Although the topics in the dataset are not as diverse as in the MTC,

the dataset is very large, containing about 1.85 million conversations with an average of 5 utterances per conversation.

We split both MTC and UDC into training, validation, and test sets, using 90%, 5%, and 5% proportions, respectively. We performed minimal preprocessing of the datasets by replacing all words except the top 50,000 most frequent words by an *UNK* symbol.

## 4.2 EVALUATION METRICS

Accurate evaluation of dialogue models is still an open challenge. In this paper, we employ both automatic and human evaluations.

### 4.2.1 AUTOMATIC EVALUATION

We employed some of the automatic evaluation metrics that are used in probabilistic language and dialogue models, and statistical machine translation. Although these metrics may not correlate well with human judgment of dialogue responses (Liu et al., 2016), they provide a good baseline for comparing dialogue model performance.

**Perplexity** - For a model with parameter $\theta$, we define perplexity as:

$$exp\left[ -\frac{1}{N_W} \sum_{k=1}^{K} log\, P_\theta(Y_1, Y_2, \ldots, Y_{N_k-1}) \right] \tag{14}$$

where $K$ is the number of conversations in the dataset, $N_k$ is the number of utterances in conversation $k$, and $N_W$ is the total number of word tokens in the entire dataset. The lower the perplexity, the better. The perplexity measures the likelihood of generating the ground truth given the model parameters. While a generative model can generate a diversity of responses, it should still assign a high probability to the ground truth utterance.

**BLEU** - The BLEU score, (Papineni et al., 2002) provides a measure of overlap between the generated response (candidate) and the ground truth (reference) using a modified n-gram precision. According to Liu et. al. (Liu et al., 2016), BLEU-2 score is fairly correlated with human judgment for non-technical dialogue (such as MTC).

**ROUGE** - The ROUGE score, (Lin, 2014) is similar to BLEU but it is recall oriented instead. It is used for automatic evaluation of text summarization and machine translation. To compliment the BLEU score, we use ROUGE-N with $N = 2$ for our evaluation.

**Distinct n-gram** - This is the fraction of unique n-grams in the generated responses. It provides a measure of diversity. Models with higher number of distinct n-grams tend to produce more diverse responses (Li et al., 2016a). For our evaluation, we use 1- and 2- grams.

**Normalized Average Sequence Length (NASL)** - This measures the average number of words in model generated responses normalized by the average number of words in the groundtruth.

### 4.2.2 HUMAN EVALUATION

For human evaluation, we follow a similar setup as Li et al. (2016a), employing crowd-sourced judges to evaluate a random selection of 200 samples. We present both the multi-turn context and the generated responses from the models to 3 judges and asked them to rank the general response quality in terms of relevance and informativeness. For $N$ models, the model with the lowest quality is assigned a score 0 and the highest is assigned a score N-1. Ties are not allowed. The scores are normalized between 0 and 1 and averaged over the total number of samples and judges.

## 4.3 BASELINE

We compare the performance of our model to (V)HRED (Serban et al., 2016; 2017b), since they are the closest to our approach in implementation and are the current state of the art in open-domain dialogue models. HRED is very similar to our proposed generator, but without the input utterance attention and noise samples. VHRED introduces a latent variable to the HRED between the $cRNN$ and the $dRNN$ and was trained using the variational lower bound on the log-likelihood.

Table 1: Generator Performance Evaluation

| Model | Teacher Forcing | | Autoregression | | | | Human |
| | Perplexity | $-logD(G(.))$ | BLEU-2 | ROUGE-2 | DISTINCT-1/2 | NASL | Evaluation |
|---|---|---|---|---|---|---|---|
| **MTC** | | | | | | | |
| HRED | 31.92/36.00 | NA | 0.0474 | 0.0384 | 0.0026/0.0056 | 0.535 | 0.2560 |
| VHRED | 42.61/44.97 | NA | 0.0606 | 0.1181 | 0.0048/0.0163 | 0.831 | 0.3909 |
| hredGAN_u | **23.57/23.54** | 23.57/23.54 | 0.0493 | 0.2416 | 0.0167/0.1306 | 0.884 | 0.5582 |
| hredGAN_w | 24.20/24.14 | **13.35/13.40** | **0.0613** | **0.3244** | **0.0179/0.1720** | **1.540** | **0.7869** |
| **UDC** | | | | | | | |
| HRED | 69.39/86.40 | NA | 0.0177 | 0.0483 | 0.0203/0.0466 | 0.892 | 0.3475 |
| VHRED | 98.50/105.20 | NA | 0.0171 | 0.0855 | 0.0297/0.0890 | 0.873 | 0.4046 |
| hredGAN_u | 56.82/57.32 | 10.09/10.08 | 0.0137 | 0.0716 | 0.0260/0.0847 | **1.379** | 0.6133 |
| hredGAN_w | **47.73/48.18** | **8.37/8.36** | **0.0216** | **0.1168** | **0.0516/0.1821** | 1.098 | **0.6905** |

The VHRED can generate multiple responses per context like hredGAN, but has no specific criteria for selecting the best response.

The HRED and VHRED models are both trained using the Theano-based implementation obtained from `https://github.com/julianser/hed-dlg-truncated`. The training and validation sets used for UDC and MTC dataset were obtained directly from the authors[1] of (V)HRED. For model comparison, we use a test set that is disjoint from the training and validation sets.

## 4.4 RESULTS

We have two variants of hredGAN based on the noise injection approach, i.e., hredGAN with utterance-level (*hredGAN_u*) and word-level (*hredGAN_w*) noise injections.

We compare the performance of these two variants with HRED and VHRED models.

**Perplexity**: The average perplexity per word performance of all the four models on MTC and UDC datasets (validation/test) are reported in the first column on Table 1. The table indicates that both variants of the hredGAN model perform better than the HRED and VHRED models in terms of the perplexity measure. However, using the adversarial loss criterion (Eq. equation 8), the hredGAN_u model performs better on MTC and worse on UDC. Note that, for this experiment, we run all models in teacher forcing mode.

**Generation Hyperparameter**: For adversarial generation, we perform a linear search for $\alpha$ between 1 and 20 at an increment of 1 using Eq. equation 13, with sample size $L = 64$, on validation sets with models run in autoregression. The optimum values of $\alpha$ for hredGAN_u and hredGAN_w for UDC are 7.0 and 9.0 respectively. The values for MTC are not convex, probably due to small size of the dataset, so we use the same $\alpha$ values as UDC. We however note that for both datasets, any integer value between 3 and 10 (inclusive) works well in practice.

**Quantitative Generator Performance**: We run autoregressive inference for all the models (using optimum $\alpha$ values for hredGAN models and selecting the best of $L = 64$ responses using a discriminator) with dialogue contexts from a unique test set. Also, we compute the average BLEU-2, ROUGE-2(f1), Distinct(1/2) and normalized average sequence length (NASL) scores for each model and summarize the results in the middle of Table 1. Distinct(1/2) largely agrees with the perplexity score. Most scores, similar to the perplexity, indicate that hredGAN models perform better than (V)HRED on both datasets. However, on the UDC and MTC, ROUGE and BLUE, respectively scores VHRED slightly better than hredGAN_u but still worse than hredGAN_w.

A good dialogue model should find the right balance between precision (BLEU) and diversity. We strongly believe that our adversarial approach is better suited to solving this problem. As hredGAN generators explore diversity, the discriminator ranking gives hredGAN an edge over (V)HRED because it helps detect responses that are out of context and the natural language structure (Table 2). Also, the ROGUE(f1) performance indicates that hredGAN_w strikes a better balance between pre-

---

[1]UDC was obtained from `http://www.iulianserban.com/Files/UbuntuDialogueCorpus.zip`, and the link to MTC was obtained privately.

Table 2: Example of Discriminator Ranking with hredGAN_w

| Item | $D(G(.))$ | Utterance |
|---|---|---|
| **MTC** | | |
| Context_0 | NA | perhaps <person> had a word with the man upstairs . |
| Context_1 | NA | a word ? i ' m sure by now he ' s engineered a hostile takeover . |
| Response_0 | 0.996 | <person> , i know what you ' re saying , <person> , that ' s not what i ' m saying . |
| Response_1 | 0.991 | <person> , i know . i was just about to help the guy . |
| Response_2 | 0.315 | <person> , i ' m sorry . |
| Response_3 | 0.203 | <person> , i ' m a little out . |
| Context_0 | NA | says he wanted food . <person> . he wanted the gold . |
| Context_1 | NA | how ' s he going to want the gold ? he couldn ' t even know we had it . |
| Response_0 | 0.998 | <person> , i know . but it ' s not him , it ' s the only way he ' s got it all figured |
| Response_1 | 0.981 | <person> , i know . but i have to tell you . these things are really stupid and you think i was wrong ? |
| Response_2 | 0.690 | <person> , i ' m sure he did . |
| Response_3 | 0.314 | <person> , i ' m not sure . |
| **UDC** | | |
| Context_0 | NA | The netboot one is suppose to download packages from the net. |
| Context_1 | NA | like the ones to be installed? or the installed to be run? |
| Response_0 | 0.993 | you don ' t need to install the whole system , just install the ubuntu installer |
| Response_1 | 0.952 | you can install the ubuntu installer from the ubuntu menu |
| Response_2 | 0.749 | I ' m not sure , I don ' t know . |
| Response_3 | 0.184 | you can |
| Response_4 | 0.003 | you a . 1 . . . . . . . . . . . . . . . . . . . . |
| Context_0 | NA | DJones: update manager won't detect 12.04.1 as a new version if you are already running 12.04, because 12.04.1 = 12.04 + lots of package updates |
| Response_0 | 0.991 | did you try a clean install of the latest version ? |
| Response_1 | 0.981 | try installing the latest _UNK and see if it works |
| Response_2 | 0.615 | I ' m not sure you have a problem . . . |
| Response_3 | 0.191 | try sudo apt-get remove the package that is not installed |
| Response_4 | 0.002 | try the _UNK . 1 . 1 . 1 . 1 . 1 . _UNK . deb |

Table 3: Sample responses of HRED, VHRED and hredGAN.

| **Movie** | |
|---|---|
| **Context_0** | perhaps <person> had a word with the man upstairs . |
| **HRED** | i ' m sorry , <person> . |
| **VHRED** | <person>< ' t do it . |
| **hredGAN_w** | <person> , i ' m not gon na be okay . i just don ' t think it ' s a good idea . |
| **hredGAN_u** | <person> , i cause it was all the way to you . |
| **Context_1** | a word ? i ' m sure by now he ' s engineered a hostile takeover . |
| **HRED** | i ' m sorry , <person> . |
| **VHRED** | <person> ' t do it . |
| **hredGAN_w** | <person> , i know what you ' re saying , <person> , that ' s not what i ' m saying . |
| **hredGAN_u** | <person> , i have to tell you i ' m not the same . |
| **Context_0** | <person> ' s . |
| **HRED** | i ' m sorry , <person> . |
| **VHRED** | <person> ' t do it . |
| **hredGAN_w** | <person> , i ' m sure it ' s the only thing i ever imagined . |
| **hredGAN_u** | <person> , i ' ll be right there . |
| **Context_1** | can i ask you something about her ? |
| **HRED** | i don ' t know . |
| **VHRED:** | you ' re not going to be here . |
| **hredGAN_w** | i don ' t know . i think i ' ve heard of it . |
| **hredGAN_u** | i ' m sorry , i can ' t . |
| **Ubuntu** | |
| **Context_0** | this is Dacs i am in i used 'sudo statx and it let me in |
| **HRED** | what 's the problem ? |
| **VHRED** | I don't know what you want to do . if you want to do it , you need to do it to do it in the terminal ? |
| **hredGAN_w** | you don ' t have to do anything , just type in the terminal and it should be to find the right device |
| **hredGAN_u** | you can have a look at the output of the command . . . |
| **Context_1** | good deal... cat /etc/X11/default-display-manager |
| **HRED** | what 's the problem ? |
| **VHRED** | do you know what you want to do ? |
| **hredGAN_w** | I ' m trying to figure a command that I can find to find out the file that I can find in the file" |
| **hredGAN_u** | I don ' t see the point , but I ' m not sure how to do that . |
| **Context_2** | /usr/sbin/lightdm http://paste.ubuntu.com/1286224/ <—- my /etc/X11/xorg.conf |
| **HRED** | what 's the problem ? |
| **VHRED** | is there a way to do that in the terminal ? |
| **hredGAN_w** | did you just type \" sudo mount -a \" ? |
| **hredGAN_u** | i have no idea , i just installed ubuntu and i have no idea how to do that |

cision (BLEU) and diversity than the rest of the models. This is also obvious from the quality of generated responses.

**Qualitative Generator Performance:** The results of the human evaluation is reported in the last column of Table 1. The human evaluation agrees largely with the automatic evaluation. The hredGAN_w performs best on both datasets although the gap is more on the MTC than on the UTC. This implies that the improvement of HRED with adversarial generation is better than with variational generation (VHRED). In addition, looking at the actual samples from the generator outputs in Table 6 shows that hredGAN especially hredGAN_w performs better than (V)HRED. While other models produce short and generic utterances, hredGAN_w mostly yields informative responses. For example, in the first dialogue in Table 6, when the speaker is sarcastic about "the man upstairs", hredGAN_w responds with the most coherent utterance with respect to the dialogue history. We see similar behavior across other samples. We also note that although hredGAN_u's responses are the longest on Ubuntu (in line with the NASL score), the responses are less informative compared to hredGAN_w resulting into a lower human evaluation score. We reckon this might be due to a mismatch between utterance-level noise and word-level discrimination or lack of capacity to capture the data distribution using single noise distribution. We hope to investigate this further in the future.

**Discriminator Performance:** Although only hredGAN uses a discriminator, the observed discriminator behavior is interesting. We observe that the discriminator score is generally reasonable with longer, more informative and more persona-related responses receiving higher scores as shown in Table 2. It worth to note that this behavior, although similar to the behavior of a human judge is learned without supervision. Moreover, the discriminator seems to have learned to assign average score to more frequent or generic responses such as "I don't know", "I'm not sure" and so on, and

high score to rearer answers. That's why we sample a modified noise distribution during inference so that the generator can produce rearer utterances that will be scored high by the discriminator.

## 5 CONCLUSION AND FUTURE WORK

In this paper, we have introduced an adversarial learning approach that addresses response diversity and control of generator outputs, using an HRED-derived generator and discriminator. The proposed system outperforms existing state-of-the-art (V)HRED models for generating responses in multi-turn dialogue with respect to automatic and human evaluations. The superiority of the adversarial generation (hredGAN) over the variational generation (VHRED) is in line with other generative models employing these approaches. Our analysis also concludes that the word-level noise injection seems to perform better in general.

While this is a good starting point, we recognize the need to explore further improvements to the proposed adversarial framework: In the future, we hope to: explore which noise level works with which discrimination level; consider a multi-resolution discriminator with combined word- and utterance-level discriminations; and explore further tuning of the generator and discriminator models.

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

APPENDIX

## 6 RELATED WORK

Our work is related to end-to-end neural network–based open domain dialogue models. Most neural dialogue models use transduction frameworks adapted from neural machine translations (Sutskever et al., 2014; Bahdanau et al., 2015). These `Seq2Seq` networks are trained end-to-end with MLE criteria using large corpora of human-to-human conversation data. Others use GAN's discriminator as a reward function in a reinforcement learning framework (Yu et al., 2017) and in conjunction with MLE (Li et al., 2017; Che et al., 2017). Zhang et al. (2017) explored the idea of GAN with a feature matching criterion. Xu et al. (2017) and Zhang et al. (2018) employed GAN with an approximate embedding layer as well as with adversarial information maximization respectively to improve `Seq2Seq`'s diversity performance.

Still, `Seq2Seq` models are limited in their ability to capture long temporal dependencies in multi-turn conversation. Although, Li et al. (2016b) attempted to optimize a pair `Seq2Seq` models for multi-turn dialogue, the multi-turn objective is only applied at inference and not used for actual model training. Hence, the introduction of HRED models (Serban et al., 2016; 2017a;b; Xing et al., 2017) for modeling dialogue response in multi-turn conversations. However, these HRED models suffer from lack of diversity since they are trained with only MLE criteria. On other hand, adversarial system has been used for evaluating open domain dialogue models (Bruni & Fernndez, 2018; Kannan & Vinyals, 2017). Our work, hredGAN is closest to the combination of HRED generation models (Serban et al., 2016) and adversarial evaluation (Kannan & Vinyals, 2017).

Table 4: Generator Performance: HRED vs. HRED+Attn

| Model | Teacher Forcing Perplexity | Autoregression | | | |
|---|---|---|---|---|---|
| | | BLEU-2 | ROUGE-2 | DISTINCT-1/2 | NASL |
| **MTC** | | | | | |
| HRED | 31.92/36.00 | 0.0474 | 0.0384 | 0.0026/0.0056 | 0.535 |
| HRED+Attn | 26.09/26.41 | 0.0425 | 0.2239 | 0.0397/0.1567 | 0.527 |
| **UDC** | | | | | |
| HRED | 69.39/86.40 | 0.0177 | 0.0483 | 0.0203/0.0466 | 0.892 |
| HRED+Attn | 50.82/51.31 | 0.0140 | 0.0720 | 0.0473/0.1262 | 0.760 |

## 7 ABLATION EXPERIMENTS

Before proposing the above adversarial learning framework for multi-turn dialogue, we carried out some experiments that are highlighted here.

### 7.1 GENERATOR:

First, we noted that by adding an additional attention memory to the HRED generator, we improved the test set perplexity score by more than 8 and 20 points on the MTC and UDC respectively as shown in Table 4. The addition of attention also shows a strong performance at autoregressive inference across multiple metrics as well as observed improvement in response quality. Hence, the decision for the modified HRED generator.

The adversarial training, however helps to address the lack diversity observed in the generated responses.

### 7.2 DISCRIMINATOR:

Before deciding on the word-level discrimination, we experimented with utterance-level discrimination. The utterance-level discriminator trains very quickly but it leads to mostly generic responses from the generator. We also note that utterance-level discriminator scores are mostly extreme (i.e.,

either low or high). Since we had used convolutional neural network discriminator (Yu et al., 2017) in our experiments, we hope to investigate this further with other architectures.

### 7.3 ADVERSARIAL TRAINING:

Lastly, we also tried a basic policy gradient approach (Li et al., 2017), where word-level discriminator score is used as a reward for each generated word token, but this leads to training instability. This is probably due to the instability of Monte Carlo sampling over large vocabulary size. We believe this might improve with other sampling methods such as importance sampling and hope to investigate further in the future.

## 8 OTHER GENERATED SAMPLES

Table 5: Additional sample responses of HRED, VHRED and hredGAN on Movie.

| Movie | |
|---|---|
| **Context_0** | \<person\> i ' ve raised my rate . i might even cut my sessions to forty minutes . |
| **HRED** | i ' m sorry , \<person\> . |
| **VHRED** | \<person\>, \<person\>, \<person\>, \<person\> . |
| **hredGAN_w** | \<person\> , i ' m sorry , but i ' m not sure if you ' re really a very good guy . |
| **hredGAN_u** | i ' ll never see them again , \<person\> . |
| **Context_0** | i ' m going to handle this exactly the way your father would have handled this . i ' ve got it solved . |
| **Context_1** | you ' ve got it solved ? where is she ? |
| **HRED** | i don ' t know . |
| **VHRED** | \<person\>, , i ' m sorry. |
| **hredGAN_w** | to the end of a beautiful woman who will be very happy . |
| **hredGAN_u** | i ' m sorry , i just didn ' t know – |
| **Context_0** | of course not ! ¡person¿ ' s like a daughter to me . her father and i taught at university together . a good man , but he never listened . |
| **HRED** | i ' m sorry , \<person\>. |
| **VHRED** | \<person\>, mr . \<person\> . |
| **hredGAN_w** | \<person\> , i know it ' s been all six years since i told him about you . |
| **hredGAN_u** | i ' m sure you didn ' t , but i would have done a little more . |
| **Context_0** | of course not ! ¡person¿ ' s like a daughter to me . her father and i taught at university together . a good man , but he never listened . |
| **Context_1** | neither did i . |
| **HRED** | i ' m sorry , \<person\>. |
| **VHRED** | \<person\> , you ' re a \<unk\> . |
| **hredGAN_w** | \<person\>, i know you ' re think about it . think it ' ll be time , right ? |
| **hredGAN_u** | i ' ll be the first one . if it did you , you ' ll be it . |
| **Context_0** | come on , \<person\> , we could have a great week up in vermont . |
| **Context_1** | i see what you mean . but wednesday we ' ve got to be back in new york . |
| **HRED** | \<person\> . |
| **VHRED** | \<person\> , i don ' t know what you ' re going to do . |
| **hredGAN_w** | \<person\> , i know . i ' m kind of busy here . |
| **hredGAN_u** | i ' m sorry , i can ' t . |
| **Context_0** | isabel ? |
| **Context_1** | hi . |
| **HRED** | that ' s what i ' m talking about . |
| **VHRED**: | you ' re not going to be a \<unk\> , \<person\> . |
| **hredGAN_w** | \<person\> ? |
| **hredGAN_u** | what do you want to know ? |
| **Context_0** | you understand mr . \<person\> is constantly barraged with nuisance litigation and irs audits . |
| **Context_1** | of course . |
| **HRED** | that ' s right , \<person\> . |
| **VHRED**: | \<person\> , you ' re going to be a \<unk\> of the \<unk\> . |
| **hredGAN_w** | \<person\> , i ' m sure , but you ' re probably still in to the new \<person\> ' s house , huh ? |
| **hredGAN_u** | you ' re gon na be a little boy , and i ' m not gon na get out ! |
| **Context_0** | \<person\> ' t go into my room , she ' s asleep . |
| **Context_1** | \<person\> , but the nanny , mrs . \<person\> , she wants to see her . |
| **HRED** | i ' m sorry , \<person\> . |
| **VHRED**: | \<person\> , mr . \<person\> . |
| **hredGAN_w** | \<person\> , i know . but she ' s never seen me anyway . |
| **hredGAN_u** | i ' m sorry i didn ' t know what to say . |
| **Context_0** | what was that for ? |
| **Context_1** | i ' d forgotten how exciting it is working together . |
| **HRED** | i ' m sorry , \<person\> . |
| **VHRED**: | \<person\> , \<person\> . i don ' t know . |
| **hredGAN_w** | i ' m sorry , \<person\> . so many people are so interested , but you ' re a very nice guy . |
| **hredGAN_u** | i ' ve always been so sure to \<person\> . |
| **Context_0** | a teacher ' s . |
| **Context_1** | oh my god . |
| **HRED** | i ' m sorry , \<person\> . |
| **VHRED**: | \<person\> , \<person\> ' t worry about it . i don ' t know what you ' re doing . |
| **hredGAN_w** | i ' m sorry . i didn ' t mean to . i didn ' t mean to hurt you . i just wanted you |
| **hredGAN_u** | \<person\> , i don ' t want to be your friend . |
| **Context_0** | this is all we have . |
| **HRED** | \<person\> , i ' m sorry . |
| **VHRED**: | \<person\> , i don ' t know what you ' re doing . i don ' t know what i ' m going to do . |
| **hredGAN_w** | i ' m sorry , sir , but i ' m a very good man . |
| **hredGAN_u** | but you ' ve been in the apartment . |
| **Context_0** | i took the initiative . |
| **Context_1** | you haven ' t been given an assignment in this case . |
| **HRED** | \<person\> . |
| **VHRED**: | \<person\> , \<person\> , \<person\> , \<person\> . i ' m not going to get out of the \<unk\> . |
| **hredGAN_w** | i ' m not sure . \<continued_utterance\> i ' m not gon na do you this shit anyway , \<person\> . |
| **hredGAN_u** | \<person\> , i guess i didn ' t . |
| **Context_0** | \<person\> , i was just gon na leave and see if i could pick up their trail . |
| **Context_1** | no , you wait right there for me . \<person\> be on the piedmont flight tomorrow at seven . meet me at the airport . |
| **HRED** | i ' m sorry , \<person\> . |
| **VHRED**: | what do you think i ' m saying ? i don ' t know what i ' m going to do . i don ' t know what i ' m going to do . |
| **hredGAN_w** | i ' m not sure i don ' t . \<continued_utterance\> do you want to go over to the station ? |
| **hredGAN_u** | \<person\> , i don ' t know if i ' m not . |

Table 6: Additional sample responses of HRED, VHRED and hredGAN on Ubuntu.

| Ubuntu | |
|---|---|
| **Context_0** | what's the command you are running to chroot and the error message you get in return? |
| **Context_1** | i mountded the device sudo mount /dev/sda1 /mnt - that works - then sudo chroot /mnt – error is no such file or command |
| **HRED** | what 's the problem ? |
| **VHRED** | is there a way to do that ? what do you mean ? |
| **hredGAN_w** | I have no idea , I just did a sudo fdisk and it said it was mounted |
| **hredGAN_u** | I ' m trying to get the latest version of ubuntu , but I ' m getting a lot of problems . |
| **Context_0** | Hmm, have you checked that ICMP is being allowed all the way through and that the server responds to other ICMP requests? |
| **Context_1** | now i can ping it (i had to enable the lzo compression on client) i cannot reach the remote subnet adding a rule in the firewall forward the lan to vpn |
| **HRED** | what 's the problem ? |
| **VHRED** | I don't know what you want to do |
| **hredGAN_w** | I have no idea , I ' m just trying to get the ip address from the router to the router" |
| **hredGAN_u** | I ' m not sure if that ' s the case , but I ' m not sure if it ' s a good idea to have a look at the output . |
| **Context_0** | The netboot one is suppose to download packages from the net. |
| **HRED** | I don't know what you want to do . |
| **VHRED** | how do I do that ? |
| **hredGAN_w** | I ' m not sure of a command to do so I can just install the ubuntu . iso and burn a new iso |
| **hredGAN_u** | you can ' t have a look at the man page . . . |
| **Context_0** | The netboot one is suppose to download packages from the net. |
| **Context_1** | like the ones to be installed? or the installed to be run? |
| **HRED** | I don't know what the problem is . |
| **VHRED** | is there a way to install the package manager ? |
| **hredGAN_w** | you don ' t need to install the whole system , just install the ubuntu installer |
| **hredGAN_u** | I ' m trying to install a _UNK version of ubuntu , but I ' m trying to install it on my laptop . |
| **Context_0** | You trying to mount it open with the password? |
| **HRED** | I have to do that , but I have a problem with the same problem . |
| **VHRED** | is there a way to do ? |
| **hredGAN_w** | yes , I ' m not sure if it ' s mounted or not |
| **hredGAN_u** | I ' m trying to install the latest version of Ubuntu , but I ' m running on a _UNK . |
| **Context_0** | are you using lvm for your root partition? |
| **HRED** | yes , but I have a problem with a new user , but I don't want to do that . |
| **VHRED** | what is the best way to do that ? what is that ? |
| **hredGAN_w** | yes , I ' m on a live system , I ' m trying to use the partition to boot from windows |
| **hredGAN_u** | I ' m not sure if it ' s a good idea to have a backup of the system . . . |
| **Context_0** | are you using lvm for your root partition? |
| **Context_1** | i have the same config working on a 10.04 box. It appears in 12.04 lvm is on upstart and multipath is not. i do use lvm for my root folder nifty, is there any way to work around it? have you heard of anyone posting a bug about this to have it looked at? darn.... well i hope they can fix this eventually i hope they keep initrd around and move everything to upstart. Upstart is not easy to setup for programs that don't come packaged for upstart. |
| **HRED** | I don't know what the problem is , but I don't know what the problem is . ? |
| **VHRED** | is there a way to do that ? or **unknown** ? |
| **hredGAN_w** | you can also try the live cd and see if you can get it back up the live" |
| **hredGAN_u** | I ' m trying to get my wireless working on my laptop , but I ' m getting to the point that I can be able to use the wireless card to get the wireless working . |
| **Context_0** | are you using lvm for your root partition? |
| **Context_1** | i have the same config working on a 10.04 box. It appears in 12.04 lvm is on upstart and multipath is not. i do use lvm for my root folder nifty, is there any way to work around it? have you heard of anyone posting a bug about this to have it looked at? darn.... well i hope they can fix this eventually i hope they keep initrd around and move everything to upstart. Upstart is not easy to setup for programs that don't come packaged for upstart. |
| **Context_2** | yes, because i cant supply it at boot time, i want the truecrypt drive to come up by itself without manual intervention |
| **HRED** | what is the problem ? |
| **VHRED** | what do you mean ? |
| **hredGAN_w** | you can also mount a new one and put the mount command to the mount point" |
| **hredGAN_u** | I ' m trying to get my sound working , but I ' m trying to get my sound working . |
| **Context_0** | are you using lvm for your root partition? |
| **Context_1** | i have the same config working on a 10.04 box. It appears in 12.04 lvm is on upstart and multipath is not. i do use lvm for my root folder nifty, is there any way to work around it? have you heard of anyone posting a bug about this to have it looked at? darn.... well i hope they can fix this eventually i hope they keep initrd around and move everything to upstart. Upstart is not easy to setup for programs that don't come packaged for upstart. |
| **Context_2** | yes, because i cant supply it at boot time, i want the truecrypt drive to come up by itself without manual intervention |
| **Context_3** | Kinda defeats the use of it anyone could get in don't you think? |
| **HRED** | what is the problem ? |
| **VHRED** | is there a way to mount the file ? if you want to do it ? |
| **hredGAN_w** | I have no idea , I just want to get the data from the other computer |
| **hredGAN_u** | I ' m trying to get the latest driver from the nvidia driver , but I ' m trying to get the nvidia driver working |

