# OpenReview forum: "Multi-turn Dialogue Response Generation in an Adversarial Learning Framework"
_ICLR.cc/2019/Conference_

### Official Review · AnonReviewer1 · 2018-11-05
**Overall, the proposed model seems like sound and thoughtful approach, but the lack of novelty over the existing literature is a weakness.**

**Rating:** 5
**Confidence:** 5

**Review:**

This paper propose a new approach to dialogue modeling by introducing two
innovations over an established dialogue model: the HRED (Hierarchical
Recurrent Encoder-Decoder) network. The innovations are: (1) adding a GAN
objective to the standard MLE objective of the HRED model; and (2)
modifying the HRED model to include an attention mechanism over the local
conditioning information (i.e. the "call" before the present "response").

Writing: The writing was mostly ok, though there were some issues early in
Section 2. The authors rather awkwardly transition from a mathematical
formalism that included the two halves of the dialogue as X (call) and Y
(response), to a formalism that only considers a single sequence X.

Novelty and Impact:  The proposed approach explicitly combines an established
model with two components that are themselves well-established.
It's fair to say that the novelty is relatively weak. The model development
is sensible, but reasonably straightforward. It isn't clear to me that a
careful reader of the literature in this area (particularly the GAN for
text literature) will learn that much from this paper.

Experiments: Overall the empirical evaluation shows fairly convincingly
that the proposed model is effective. I do wonder why would the hredGAN
model outperform the hred model on perplexity. The hred model is
directly optimizing MLE which is directly related to the perplexity
measure, while the hredGAN include an additional objective that should
(perhaps) sacrifice likelihood. This puzzling result was not discussed and
really should be.

The generated responses, given in table 3 -- while showing some improvement
over hred and Vhred (esp. in terms of response length and specificity) --
do not fit the context particularly well. This really just shows we still
have some way to go before this challenging task is solved.

It would be useful if the authors could run an ablation study to help
resolve the relative contributions of the two innovations (GAN and
attention) to the improvements in results. Perhaps the improvement in
perplexity (discussed above) is do to the use of attention.

Detailed comments / questions

- In the paragraph between Eqns 2 and 3, the authors seem to suggest that
  teacher forcing is an added heuristic -- however this is just the
  correct evaluation of the MLE objective.

- In discussing the combined MLE-GAN objective in Eqn. 8 Does the MLE
  objective use teacher forcing? Some earlier text (discussed above) leads
  me to suspect that it does not.

---

> ### Author Response · Authors · 2018-11-09
> **Explaining novelty while awaiting human evaluation and HRED+Attention results**
>
> Thank you for your review.
>
> -- The authors rather awkwardly transition from a mathematical formalism that included the two halves of the dialogue as X (call) and Y (response), to a formalism that only considers a single sequence X.
> We try to depict the original problem by Eq.(2) which is difficult to learn or train and substitute with a well known approximation (teacher forcing) in Eq.(3). This a well established trick for training machine translation and dialogue response generation.
>
> --The model development is sensible, but reasonably straightforward.
> While HRED and GAN are will established concept, it is not trivial to combine due to the lack of end-to-end differentiability between the HRED generator and the GAN discriminator. Also, the questions of where to inject noise and how to apply discrimination also arise. Our paper addresses these problems by
> (i) proposing shared encoder and word embedding between the generator and the discriminator. Existing works with seq2seq generator either use policy gradient (Li at. Al, 2016) with no end-to-end differentiability or approximate embedding layer (Xu et al. 2017, Zhang et al. 2018) which is memory and computationally intensive with large vocabulary size.
> (ii) exploring noise injection at the word and utterance levels and discrimination at word level with RNN and at the utterance with CNN
>
> -- I do wonder why would the hredGAN model outperform the hred model on perplexity. Perhaps the improvement in perplexity (discussed above) is do to the use of attention.
> We agree with your observation but due to space limitations, we only discussed this in the Ablation experiments in the Appendix. Indeed, the presence of attention in the model is responsible for the low perplexity but it didn’t address the lack of diversity until we introduced GAN. We will include the complete results for HRED+Attention in the ablation section for comparison in the final version.
>
> -- In the paragraph between Eqns 2 and 3, the authors seem to suggest that teacher forcing is an added heuristic
> We agree with the reviewer that teacher forcing is the actual evaluation of MLE objective. However, the problem of dialogue response generation is indeed autoregressive and not teacher forcing  (Please see Lamb et. al 2016 for details). During inference, Eq. (2) is used while Eq.(3) is used as a tractable and accurate approximation during training. This discrepancy, known as exposure bias has been the subject of several dialogue modeling papers (including Lamb et. al 2016 and references there in).
> -- In discussing the combined MLE-GAN objective in Eqn. 8 Does the MLE objective use teacher forcing?
> Yes, the MLE objective uses teacher forcing as can be seen in Eq.(7).
>
> Let us know if you have additional questions while we collate and analyze the human evaluation as well as the HRED+Attention results.

---

### Official Review · AnonReviewer2 · 2018-11-06
**Interesting idea but better evaluation needed**

**Rating:** 6
**Confidence:** 4

**Review:**

This paper presents an adversarial learning model for generating diverse responses for dialogue systems based on HRED (hierarchical recurrent encoder-decoder) network. The contribution of the work mainly lies in: 1. adversarial learning, and 2. injecting Gaussian noise at word-level and sentence-level to encourage the diversity. Overall, the idea is interesting, and the automatic evaluation based on perplexity, BLEU, ROUGE, etc shows that the proposed methods outperform existing methods.

Several suggestions:
- It seems like injection noise at word-level almost always outperforms adding sentence-level noise. It would be better if the authors can explain why this happens and whether it can be applied for other response generation tasks.

- Built on above comment, the authors can also experiment with other response generation datasets, e.g. interactions on social media.

- From examples in Table 3 and 4, the generated responses are of low quality overall. I suggest the authors run human evaluation to see whether there is any significant difference among system responses by different models on aspects of informativeness and fluency at least.

---

> ### Author Response · Authors · 2018-11-09
> **Awaiting Human Evaluation**
>
> Thank you for your review.
>
> -- Why noise injection at the word level seems to performs better.
> We believe this is because the response distribution is a factor over independent word-level conditional distributions. Therefore, matching the singleton categorical distribution to the Gaussian distribution easily captures the modalities in the response space. Injecting at the utterance level seems to indicate a single probability distribution for all responses. This result is in contrast with the assumption in VHRED where noise sample was injected at the utterance level only but we believe it is responsible for the better performance.
> --Other social media data
> We will explore this in our future work
> -- Human evaluation
> We have crowdsourced the human evaluation and we will be reporting the results here soon.
>
> Let us know if you have additional questions while we collate and analyze the human evaluation results.

---

### Official Review · AnonReviewer3 · 2018-11-06
**Reasonable approach but somewhat incremental; weak evaluation setup**

**Rating:** 4
**Confidence:** 4

**Review:**


The paper applies conditional GAN to the HRED model [Serban et al., 2016] for dialogue response generation, showing improvements in terms of informativeness and diversity compared to HRED and VHRED [Serban et al., 2017].

The paper is technically solid and relatively easy to follow and the results are good, but comparisons with previous work (descriptive and experimental) are rather weak.

- Related work is incomplete: The paper specifically argues for the use of GAN to improve diversity in dialogue response generation, but this is not the first paper to do so. [Xu et al., 2017] presents a GAN-like setup that targets exactly the same goal, but that work is not cited in the paper. Same for [Zhang et al., 2018], but the latter work is rather recent (it still should probably be cited).

- Evaluation: There is no evaluation against Xu et al., which targets the same goal. The authors didn’t even compare their methods against baselines used in other GAN works for diverse response generation (e.g., MMI [Xu et al.; Zhang et al.], Li et al.’s GAN approach [Xu et al.]), which makes it difficult to draw comparisons between these related methods. As opposed to these other works, the paper doesn’t offer any human evaluation.

- It would have been nice to include an LSTM or GRU baseline, as these models are still often used in practice and the VHRED paper suggests [Serban et al., 2016; Table 1] that LSTM holds up quite well against HRED (if we extrapolate the results of VHRED vs. LSTM and VHRED vs. HRED). The ablation of GAN and HRED would help us understand which of the two is more important.

In sum, the work is relatively solid, but considering how much has already been done on generating diverse responses (including 3 other papers also using GAN), I don’t think this paper is too influential. Its main weakness is the evaluation (particularly the lack of human evaluation.)

Minor comments:

- Introduction: “diversity promoting training objective but their model is for single turn conversations”.
If these were “single turns”, they wouldn’t really be called conversations; that objective has been used with 3+ turn conversations. It can actually be applied to multi-turn dialogue as with any autoegressive generative models. For example, it has been exploited that way as a baseline for multi-turn dialogue [Li et al. 2016](“Deep Reinforcement Learning for Dialogue Generation“). Note it is not a “training objective”, but only an objective function at inference time, which is a more valid reason to criticize that paper.

- “We use greedy decoding (MLE) on the first part of the objective.” Doesn’t that hurt diversity because of MLE? what about using sampling instead (maybe with temperature)?

- Algorithm 1: the P_theta_G don’t seem to match the text of section 2. h_i is in sometimes written in bold and sometimes not (see also Eq 12 for comparison.)

- End of section 2.1: There are multiple Li et al.; specify which one.

- End of section 2.2 and 2.4: extra closing parenthesis after N(0, …))

- Figures are too small to read the subscripts.

[Xu et al. 2017]: Zhen Xu, Bingquan Liu, Baoxun Wang, Sun Chengjie, Xiaolong Wang, Zhuoran Wang, and Chao Qi. Neural response generation via gan with an approximate embedding layer. EMNLP 2017.

[Zhang et al. 2018]: Zhang, Yizhe & Galley, Michel & Gao, Jianfeng & Gan, Zhe & Li, Xiujun & Brockett, Chris & Dolan, Bill. (2018). Generating Informative and Diverse Conversational Responses via Adversarial Information Maximization.

---

> ### Author Response · Authors · 2018-11-09
> **Restating contributions while waiting for human evaluation results**
>
> Thank you for your review. We will soon post a more detailed explanation and new results with human evaluation but we want to quickly address some of the other concerns raised in your review.
>
> Thank you for pointing out those additional three papers that we missed in our discussion of the previous works. We will give a detailed explanation of how our work relates to them in the final version.
>
> --Uniqueness and Influence
> As you have also pointed out, we believe that our work is very unique. We also believe that it will be very influential for future work in this area in that we specifically investigate an end-to-end adversarial learning framework  for multi-turn dialogue. First, most of the works on adversarial dialogue response generation in the literature use ses2seq generator which has limited capacity for multi-turn dialogue history encoding.  It is computationally not feasible to always re-encode the entire conversation history at each turn. Hence, the need for the HRED generator.
> Also, in our adversarial framework, our discriminator is also multi-turn and compliments the generator at each turn. Hence, the shared dialogue history encoding which is also unique to out work.
> Furthermore, to achieve end-to-end differentiation, we also share the word embedding between the generator and discriminator. As at the time of this work,  we were not aware of the approximate embedding layer in Xu et al. 2017 and Zhang et al. 2018. However, during our study (in 2017) we took a similar approach by replacing the one-hot decoder output by the softmax probability output (with no temperature) but still with shared word embedding. We however did not see any appreciable improvement in the model performance despite the huge computational burden due to the large vocabulary size. So we decided to stick with the one-hot output and only share the word embedding. The shared embedding still allows us to achieve end-to-end differentiability. This contribution is also unique to our work.
> Finally, the adversarial response scoring by our jointly trained discriminator is calibrated based on adversarial training, whereas the scores from MMI-antiLM and MMI-bidi [Li et al. (2016)] are not. Additional learning is still required during inference to properly calibrate them.
>
> --“We use greedy decoding (MLE) on the first part of the objective.” Doesn’t that hurt diversity because of MLE? what about using sampling instead (maybe with temperature)?
> It is true that MLE sampling would ordinarily hurt diversity but with noise injection, we are able to perturb the output distribution. In fact, the exploration factor, alpha helps us to control the diversity (much smoother than categorical sampling). We noted in our experiment that increasing alpha actually increases the likelihood of high scoring by the discriminator. This makes us to believe that the generator has mapped high probability samples to high probability noise samples which the discriminator in turn mapped to low discriminator score. This shows that the rearer the noise samples, the more diverse the generator samples and the higher the discriminator score. We however, also note that very high alpha values results to low discriminator scores. Looking at these responses shows that they are less grammatical even though they are obviously diverse. Hence the decision to limit the range of alpha values.
>
> --Typos
> We appreciate your pointing out some minor typos. All those are now fixed and will be included in the final version.
>
> --Additional Evaluation
> We have crowdsourced the human evaluation and we will be reporting the results here soon.
> -- Additional Previous Work
> The three additional references provided shall be added to our citation and discussed under the previous work section.
>
> Let us know if you have additional questions while we collate and analyze the human evaluation results.

---

### Official Review · AnonReviewer5 · 2018-11-12
**Reasonable approach, but lacks novelty and better evaluation**

**Rating:** 4
**Confidence:** 4

**Review:**

This paper presented a dialog response generation method using adversarial learning framework.
The generator is based on previously proposed hierarchical recurrent encoder-decoder network (HRED), and the discriminator is a bidirectional RNN.
Noise samples are introduced in generator for response generation.
They evaluated their approach on two datasets and showed mostly better results than the other systems.

The novelty of the paper is limited.
Modeling longer dialog history (beyond the current turn) is not new, this has been used in different tasks such as dialect act classification, intent classification and slot filling, response generation, etc.
The generator is based on previous HRED.
Adding noise to generate responses is somewhat new, but that doesn’t seem to be well motivated or justified.
Why adding Gaussian noise improves the diversity or informativeness of the responses is not explained.
The idea of discriminator has been widely used recently for language generation related tasks.  What is new here? Is it the word-based metric? Sharing the context and word information with generator?  It would be helpful if the authors can clarify their contribution.

Regarding using MLE to first generate multiple hypotheses in generator, how is the quality of the n-best responses?
Is there a way to measure the goodness of the responses in some kind of reranking framework, not necessarily discriminator?

The results in the table showed the proposed method outperforms the others in terms of those objective metrics. I feel some subjective evaluations are needed to strengthen the paper.
From the samples responses in the table, it doesn’t look like the new method generates very good responses.


Detailed comments:
- Sec 2, before 2.1, last paragraph, “With the GAN objective, we can match the noise distribution, P(Z_i) to the distribution of the ground truth response, P(X_i+1|X_i).  This needs clarification.
- Figure 1: caption, “left” and “right” are misplaced.
- sec 2.1, last paragraph, without Z_i, the net could still learn a mapping from X_i to Y_i, but would produce deterministic outputs.  I think the authors mean that the system generates a probability distribution P(Y_i|X), the output is the most likely one from that. However, if the output is a distribution, the system can also do some sampling and not necessarily output the top one.  This is not that different from adding noise in the history — if that’s based on some distribution, then it may still be deterministic.

---

> ### Author Response · Authors · 2018-11-19
> **Novelty and Contribution**
>
> Thank you for your review. We will soon post a more detailed explanation and new results with human evaluation but we want to quickly address some of the other concerns raised in your review.
>
> Novelty and Significance:
>
> Our work employs a similar approach as the VHRED which applies variational approach to improve the HRED. One could argue that HRED and variational bayes already exists independently, but VHRED shows the impact of variational training on the HRED generator. In the same vein, we also examine the impact of adversarial training on the HRED generator since both the variational and adversarial approaches are very competitive for training generative models. That's why we compare hredGAN, VHRED and HRED in our evaluation. In summary, our work shows that our proposed adversarial training performs better than the variational training in VHRED.
>
> Our other contributions relate more with the actualization of the adversarial training. To achieve an end-to-end gradient flow from the discriminator to the generator, we share the word embedding and context information between the generator and discriminator as you have rightly mentioned. The alternative would be either using a policy gradient method (REINFORCE) or passing the softmax output instead of the argmax output to the discriminator, both of which did not perform better than the embedding sharing in our experiments.   On the choice of the discriminator, we found the aggregated word-level metric to be much better than the utterance-level metric so we propose a word-level metric as you have rightly mentioned.
>
> Detailed Comments:
> We have corrected the misplaced caption in Fig. 1.
>
> Your explanation of "sec 2.1, last paragraph" is spot on. We opted to sample a latent and much smoother noise distribution, P(Z_i) while keeping the output as the most likely from P(Y_i|X, Z_i) instead of sampling P(Y_i|X) discretely. Whereas the discrete sampling of  P(Y_i|X) does not work well in practice due the large vocabulary size, our latent sampling, P(Y_i|X, Z_i)P(Z_i) allows us to control the diversity of the output from the variation of the noise sample via the parameter \alpha. Therefore, we are able to generate a much more coherent samples than possible with discrete sampling of P(Y_i|X).
>
> Let us know if you have additional questions as we collate and analyze the human evaluation results. We will be posting the new results in the next few days.

---

### Author Response · Authors · 2018-11-27
**Human evaluation results added**

Thank you for your reviews. We have conducted human evaluation of the models presented in the paper and added the results with some discussion to the updated paper. We also expanded the previous work section with additional citations and added more detailed results to the ablation studies to show the impact of local attention. We have updated the paper with these and other changes requested/suggested by the reviewers. Please endeavor to review them and let us know if you have additional questions or concerns.

---

### Meta-Review · Area_Chair1 · 2018-12-13
**Approach being incremental is a consistent concern amongst reviewers**

**Confidence:** 4
**Recommendation:** Reject

**Metareview:**


This paper proposes an adversarial learning framework for dialogue generation. The generator is based on previously proposed hierarchical recurrent encoder-decoder network (HRED) by Serban et al., and the discriminator is a bidirectional RNN. Noise is introduced in generator for response generation.
The approach is evaluated on two commonly used corpora, movie data and ubuntu corpus.

In the original version of the paper, human evaluation was missing, an issue raised by all reviewers, however, this has been added in the revisions. These supplement the previous automated measures in demonstrating the benefits and significant gains from the proposed approach.

All reviewers raise the issue of the work being incremental and not novel enough given the previous work in HRED/VHRED and use of hierarchical approaches to model dialogue context. Furthermore, noise generation seems new, but is not well motivated, justified and analyzed.